# Diagnostic Accuracy of Differential-Diagnosis Lists Generated by Generative Pretrained Transformer 3 Chatbot for Clinical Vignettes with Common Chief Complaints: A Pilot Study

**DOI:** 10.3390/ijerph20043378

**Published:** 2023-02-15

**Authors:** Takanobu Hirosawa, Yukinori Harada, Masashi Yokose, Tetsu Sakamoto, Ren Kawamura, Taro Shimizu

**Affiliations:** Department of Diagnostic and Generalist Medicine, Dokkyo Medical University, Tochigi 321-0293, Japan

**Keywords:** artificial intelligence, generative pretrained transformers, diagnostic accuracy, diagnosis, clinical decision support, AI chatbot, natural language processing

## Abstract

The diagnostic accuracy of differential diagnoses generated by artificial intelligence (AI) chatbots, including the generative pretrained transformer 3 (GPT-3) chatbot (ChatGPT-3) is unknown. This study evaluated the accuracy of differential-diagnosis lists generated by ChatGPT-3 for clinical vignettes with common chief complaints. General internal medicine physicians created clinical cases, correct diagnoses, and five differential diagnoses for ten common chief complaints. The rate of correct diagnosis by ChatGPT-3 within the ten differential-diagnosis lists was 28/30 (93.3%). The rate of correct diagnosis by physicians was still superior to that by ChatGPT-3 within the five differential-diagnosis lists (98.3% vs. 83.3%, *p* = 0.03). The rate of correct diagnosis by physicians was also superior to that by ChatGPT-3 in the top diagnosis (53.3% vs. 93.3%, *p* < 0.001). The rate of consistent differential diagnoses among physicians within the ten differential-diagnosis lists generated by ChatGPT-3 was 62/88 (70.5%). In summary, this study demonstrates the high diagnostic accuracy of differential-diagnosis lists generated by ChatGPT-3 for clinical cases with common chief complaints. This suggests that AI chatbots such as ChatGPT-3 can generate a well-differentiated diagnosis list for common chief complaints. However, the order of these lists can be improved in the future.

## 1. Introduction

Artificial intelligence (AI), a simulation of human intelligence into machines, has already assumed an essential role in daily life, including healthcare. Natural language processing (NLP) is a field of AI and plays a key role in clinical decision support (CDS) systems [1]. One of the most notable examples of such NLP models is the Generative Pretrained Transformer (GPT) model [1].

### 1.1. Clinical Decision Support and Artificial Intelligence

CDS systems that use AI can perform a diagnosis and provide physicians with treatment suggestions [2]. They improve diagnostic accuracy, efficiency, and safety through optimal interactions between physicians and CDS systems.

AI chatbots have recently been developed and refined for multiple fields [3], including healthcare [4,5,6,7]. For example, CDS systems that can answer questions have emerged [8,9,10]. Particularly in clinical medicine, the usefulness of intelligent triage systems has also been demonstrated during the coronavirus disease 2019 (COVID-19) pandemic [7,11]. Additionally, NLP systems that extract disease symptoms from clinical texts have been developed [12,13,14,15].

### 1.2. GPT-3 and ChatGPT-3

Generative Pretrained Transformer-3 (GPT-3) is a third-generation generative pretrained transformer model [16]. It is an autoregressive language model, which is a type of statistical language model that predicts the probability of a word in a text sequence. It was trained using 175 billion parameters. In GPT-3, autoregressive language models predict the next element in a text, based on previous natural-language texts. GPT-3 was designed as a language generation model that adapts to answer not only yes/no questions but also complex ones [7]. GPT-3 has been trained on several major data sources, such as Common Crawl, WebText2, Books1, Books2, and English-language Wikipedia [17]. Some of these data sources, including the Common Crawl [18] and English-language Wikipedia [19], contain not only general information but also science, medicine, and health information. 

ChatGPT-3, an AI chatbot developed by the OpenAI foundation, is an application based on the latest version of GPT-3. It can create surprisingly intelligent-sounding texts in response to user queries. ChatGPT-3 has achieved remarkable success in question-answer tasks. The ChatGPT-3 model was trained using reinforcement learning based on human feedback [20]. After its release, ChatGPT-3 has attracted an increasing number of users. Although most users have positively reacted over social media [21], some are concerned about its negative impact from educational aspects [22].

### 1.3. Other CDS Systems

Recently, several CDS systems, such as web-based symptom checkers [23] and differential-diagnosis generators [24,25] have been developed. Generally, a symptom checker is intended for use by the general public, whereas a differential-diagnosis generator is intended for use by healthcare providers. CDS systems are digital tools used to assess the potential causes of medical complaints. Therefore, they are becoming increasingly popular. Unfortunately, although the diagnostic accuracy of CDS systems has improved, it is still poor [23,25,26,27]. For the symptom checkers, a review article demonstrated that the rates of correct diagnoses were 71.1% and 45.5%, within the ten differential-diagnosis lists and in the top diagnosis, respectively [23]. The rate of correct diagnosis within the five differential-diagnosis lists generated by symptom checkers is 51% [27]. For the differential diagnoses generator, a review article demonstrated the rates of correct diagnoses were 44–78% and 23–28%, within the ten differential-diagnosis lists and in the top diagnosis, respectively [25]. Additionally, the input styles of some CDS systems constitute divided fields, including multiple text fields, pull-down style, and checkbox style, which are not considered free text writing.

Compared with previous CDS systems, ChatGPT-3 including the hyperparameters was not specifically tuned for medical diagnosis. For general users, the interface of ChatGPT-3 is user-friendly with free text input and sentence output. 

### 1.4. Related GPT-3 Work for Healthcare

GPT-3 is a the relatively new method. Therefore, medical research articles related to it are limited. Some GPT-3-related medical articles have been reported about a medical dialogue summarizer [28], navigation application for electronic health record systems [5], and its implementation in clinical use [7] including ophthalmology [4] and dementia prediction [29].

The diagnostic accuracy of the GPT-3 model is considerably limited. A preprint article revealed the correct diagnosis to be 88% within the three differential-diagnosis lists [30]. Therefore, the diagnostic accuracy of the differential-diagnosis lists generated by AI chatbots, including ChatGPT-3, is unknown.

To evaluate the diagnostic accuracy of AI chatbots, we hypothesized lists of common clinical vignettes with good diagnostic accuracy. We prepared the clinical vignettes with common chief complaints created by physicians. Subsequently, ChatGPT-3 generated the differential-diagnosis for the clinical vignettes. Then, we evaluated the diagnostic accuracy compared with physicians’ diagnosis. Finally, we will discuss the scope for its future study.

## 2. Materials and Methods

This materials and methods section presents the study design, case materials, differential-diagnosis lists generated by ChatGPT-3, measurements and definitions, sample size calculation, and analysis.

### 2.1. Study Design

We investigated the diagnostic accuracy of the differential-diagnosis lists generated by ChatGPT-3 for clinical cases with common chief complaints. The study was conducted at the General Internal Medicine (GIM) Department (Department of Diagnostic and Generalist Medicine) of Dokkyo Medical University Hospital, Shimotsuga, Tochigi, Japan. Because the study used case vignettes, approval by the Ethics Committee and requirement for individual consent were not required.

### 2.2. Case Materials

This study used 30 written clinical vignettes that included only mock patient history and physical examination with vital signs. We selected the chief complaints from a previous study [31]. After excluding non-differential complaints (“cuts and lacerations” and “motor vehicle accident”), we selected the following top ten chief complaints: abdominal pain, fever, chest pain, breathing difficulty, joint pain, vomiting, ataxia/difficulty walking, back pain, cough, and dizziness. All clinical vignettes and single correct answers for each vignette were generated by three individual GIM physicians (MY, TS, and RK). Each clinical course included a typical presentation and was expected to be correctly answered by medical students or junior residents. Three co-researchers created clinical vignettes for ten common chief complaints in English, totaling 30 vignettes. Five differential diagnoses for each case were independently created by two co-researchers who did not create clinical vignettes. When the two independent co-researchers made the same differential diagnoses for each vignette, the diagnoses were defined as consistent differential diagnoses. The consistent differential diagnoses were determined by another GIM physician (YH). Online Appendix A shows a sample vignette, a single correct diagnosis, and five differential diagnoses. The chief complaint of the sample vignette was not included in the study.

### 2.3. Differential-Diagnosis Lists Generated by ChatGPT-3

We used the ChatGPT-3 Dec 15 Version (ChatGPT 3.5, https://chat.openai.com/chat, OpenAI, L.L.C., San Francisco, CA, USA) on 5 January 2023. The system was initially released as free to the general public on 30 November 2022. We did not tune any hyperparameters of ChatGPT-3. The main investigator (TH) typed the following text in the ChatGPT-3 prompt: “Tell me the top ten suspected illnesses for the following symptoms: (copy & paste each clinical vignette)”. The order of the clinical vignettes presented to ChatGPT-3 was randomly generated using a computer-generated order table. Additionally, TH cleared the previous conversation before typing the input text in the prompt and generating the differential-diagnosis list. This was because ChatGPT-3 tends to reinforce previous conversations in the same chat. Initial answers were used as the ten differential-diagnosis lists generated by ChatGPT-3. TH was not aware of the single correct answer and the five differential-diagnosis lists created by the co-researchers. Another co-researcher (YH) assessed the primary and secondary outcomes. The study design is shown in Figure 1. Figure 2 shows an example of a differential-diagnosis list generated by ChatGPT-3.

### 2.4. Measurements and Definitions

The measurement was the total rate of correct diagnoses within the ten differential-diagnosis lists, five differential-diagnosis, and top diagnoses generated by ChatGPT-3. These measures adopted a binary approach, wherein the correct presence of a diagnosis on any list was scored as one and its absence as zero.

As an exploratory analysis, we also measured the rate of consistent differential diagnoses made by two physicians within the ten differential-diagnosis lists generated using ChatGPT-3. The number of consistent differential diagnoses between the two physicians were used as the denominator, and the number of the diagnoses within the ten differential-diagnosis lists generated by ChatGPT-3 was used as the numerator.

### 2.5. Sample Size

To determine the required sample size, we set the α and β errors to 0.05 and 0.2, respectively. The expected total score of the physicians was set to 99%, and that of ChatGPT-3 was set to 90%. A total of 25 clinical vignettes were required. The sample size was ensured by including 10 common chief complaints generated by three independent physicians using 30 clinical vignettes.

### 2.6. Analysis

Categorical or binary variables are presented as numbers (percentages) and were compared using the chi-squared test. G*power version 3.1.9.6 (Department of Psychology of Heinrich Heine University, Düsseldorf, Germany) was used for the statistical power analysis. Statistical significance was defined as a *p*-value < 0.05. R 4.2.2 (The R Foundation for Statistical Computing, Vienna, Austria) was used for all analyses, except for the statistical power analysis.

## 3. Results

Thirty clinical vignettes for ten common complaints were evaluated in this study.

The total rate of correct diagnoses within the ten differential-diagnosis lists generated by ChatGPT-3 was 28/30 (93.3%); Table 1 presents the results. Online Appendix A shows a single correct diagnosis of the clinical vignettes and ten differential-diagnosis lists created via ChatGPT-3.

The total rate of correct diagnosis within the five differential-diagnosis lists generated by ChatGPT-3 was 25/30 (83.3%). The rate of correct diagnosis made by physicians was still superior to that within the five differential-diagnosis lists generated by ChatGPT-3 (98.3% vs. 83.3%, *p* = 0.03). The total rate of correct diagnoses as the top diagnosis made by physicians was still superior to that generated by ChatGPT-3 (53.3% vs. 93.3%, *p* < 0.001). Additionally, the rate of correct diagnoses as the top diagnosis made by physicians about the vomiting cases was still superior to that generated by ChatGPT-3 (0% vs. 100%, *p* = 0.02). Table 1 presents the results.

As an exploratory analysis, we measured the rate of consistent differential diagnoses between the two physicians, which was 88/150 (58.7%). In total, 88 of consistent differential diagnoses between the two physicians were used as the denominator and 62 of the diagnoses within the ten differential-diagnosis lists generated by ChatGPT-3 was used as the numerator. The rate of consistent differential diagnoses by ChatGPT-3 within the ten lists was 62/88 (70.5%), as presented in Table 2.

Post-hoc power analysis indicated that the statistical power for comparison of the total score within the five differential-diagnosis lists was 0.49. The statistical power for the comparison of the total score in the top diagnosis was 0.94.

## 4. Discussion

This section presents the principal findings, strengths, limitations, risk for general user, and comparison with previous work.

### 4.1. Principal Findings

This pilot study presents several main findings.

To the best of our knowledge, this is the first pilot study to evaluate the diagnostic accuracy of differential-diagnosis lists generated by ChatGPT-3 for clinical vignettes with common chief complaints. ChatGPT-3 can generate differential-diagnosis lists for common clinical vignettes with good diagnostic accuracy. The total rate of correct diagnosis was more than 90% for the ten differential-diagnosis lists generated by ChatGPT-3. Furthermore, the total rate of correct diagnoses was more than 80% for the five differential-diagnosis lists generated by ChatGPT-3. The total rate of correct diagnoses as the top diagnosis generated by ChatGPT-3 was more than 50%. However, the correct diagnosis rate of physicians was superior to that of ChatGPT-3. These results suggest that ChatGPT-3 can generate differential-diagnosis lists with good diagnostic accuracy. However, in the future, the order of the lists can be improved via hyperparameter tuning with specific training for eHealth diagnosis.

Additionally, consistent differential diagnoses made by two independent physicians were suggested to be important differential diagnoses, which were collective intelligence to reduce misdiagnoses [32]. These diagnoses should be included in the list generated by ChatGPT-3. The rate of the consistent differential diagnoses within the ten differential-diagnosis lists by ChatGPT-3 was >70%. This means that physicians should know that approximately 30% of important differential diagnoses would be owing to the lack of differential-diagnosis lists generated by ChatGPT-3.

### 4.2. Strengths

The strengths of this study include the use of ChatGPT-3. This suggests that some AI chatbots developed for routine use can generate differential diagnoses with good diagnostic accuracy for common clinical vignettes. Additionally, the user interface of ChatGPT-3 is friendly for general users owing to its free text writing style and text style output. Users can change the input style to numbers or words and the input volume with or without additional medical information. They can also change the output style, including the setting, number of differential diagnoses, and reasons for their preferences. Another strength of this study was that the GIM physicians diagnosed a variety of symptoms, including common symptoms, and prepared the case materials.

### 4.3. Limitations

This study and ChatGPT-3 have several limitations.

The most significant limitation of this study is that it was vignette-based instead of being based on real patients’ cases. This is because ChatGPT-3 is not approved for clinical use to obtain health information. In the future, after tuning for diagnosis and compliance with the Health Insurance Portability and Accountability Act [7], the clinical use of AI chatbots will be further evaluated. Another limitation is the content validation. Almost all the chief complaints had a variety of final diagnoses. However, some of them, such as dizziness, were similarly diagnosed among the vignettes. Clinical vignettes were peer reviewed by each research member to confirm the content validity. Additionally, we did not evaluate the differential diagnoses as emergency or clinically important like the severity or rarity. Therefore, future research on the evaluation of differential-diagnosis lists, including clinically important diseases, is required.

The main limitations of ChatGPT-3 are that it can be misleading, biased, not transparent, and lacking recent knowledge. The risk of being misleading can result in poor medical decisions and harm to patients [33]. After each update, the number of misleading responses was reduced [34]. However, users, including both healthcare providers and patients, should always perform external validation. AI chatbots, including ChatGPT-3, are trained on databases created from human writing. Therefore, they contain several biases [7,16,35,36]. For example, gender [37,38], racial [17], and religious [17] biases have been reported. In ChatGPT-3, the entire algorithm has not been made available to the public. Therefore, how the hyperparameters of ChatGPT-3 were tuned for medical diagnosis was not clarified. Owing to the lack of transparency, efficient and accurate auditing algorithms and systems are needed [16]. AI chatbots are only as knowledgeable as the data they are trained on. ChatGPT-3 has been trained on data until 2020. Moreover, it is also not connected to the internet. This implies that ChatGPT-3 cannot answer based on recent knowledge. Physicians need to check the answers generated by ChatGPT-3 to update them based on recent knowledge, with external validation.

Despite these risks, the next step is to validate our results in more complex clinical cases and to further tune the model to clinical medicine and diagnosis.

### 4.4. Risk for General User

In this study, physicians developed the clinical vignettes. However, the risk of change in diagnostic accuracy owing to incomplete medical information exists for a general user. It is difficult for a general user to input an optimal medical term. The impact of the incompleteness of medical information on the output of the AI chatbot will be studied in the future.

### 4.5. Comparison with Prior Work

Compared with previous research on CDS systems, including symptom checkers and differential-diagnosis generators, the diagnostic accuracy of the differential-diagnosis lists generated by ChatGPT-3 was good.

Compared with a previous review of symptom checkers [23], the rate of correct diagnoses within the ten differential-diagnosis lists generated by ChatGPT-3 was more than 20% (93.3% vs. 71.1%). The total rate of correct diagnoses as the top diagnosis generated by ChatGPT-3 was also high (53.3% vs. 45.5%). Furthermore, compared with previous research on symptom checkers [27], the rate of correct diagnosis within the five differential-diagnosis lists generated by ChatGPT-3 was more than 30% (83.3% vs. 51.0%).

Compared with the previous review of the differential diagnoses generator [25], the rate of correct diagnoses within the ten differential-diagnosis lists generated by ChatGPT-3 was more than 15% (93.3% vs. 44–78%). The total rate of correct diagnosis as the top diagnosis generated by the ChatGPT-3 was more than 25% (53.3% vs. 23–28%).

This inconsistency was partly due to a difference in the study designs, including case materials and systems. These results were difficult to interpret because they were not directly evaluated. Therefore, in future research, direct comparisons between ChatGPT-3 and other CDS systems are required.

## 5. Conclusions

This study demonstrated the diagnostic accuracy of the differential-diagnosis generated by ChatGPT-3 for clinical vignettes with common chief complaints. The total rate of the correct diagnoses within ten differential-diagnosis lists generated by ChatGP-3 was more than 90%. This suggests that not only specific systems developed for diagnosis but also general AI chatbots, such as ChatGPT-3, can generate well-differentiated diagnosis lists for common chief complaints. However, the order of these lists can be improved in the future. Future studies should focus on evaluating more difficult and complex cases with the development of well-trained AI chatbots for diagnoses. Additionally, an optimal collaboration among physicians, patients, and AI should also be evaluated in eHealth.

## Figures and Tables

**Figure 1 ijerph-20-03378-f001:**
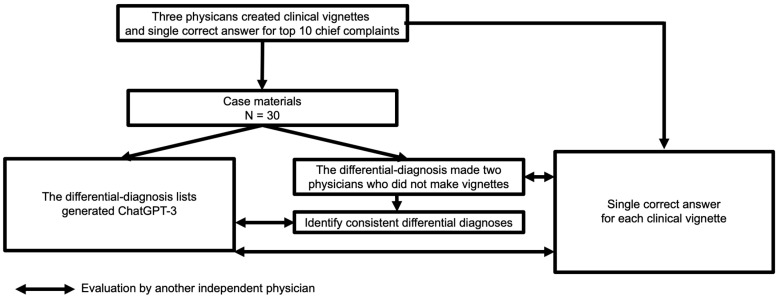
Study design.

**Figure 2 ijerph-20-03378-f002:**
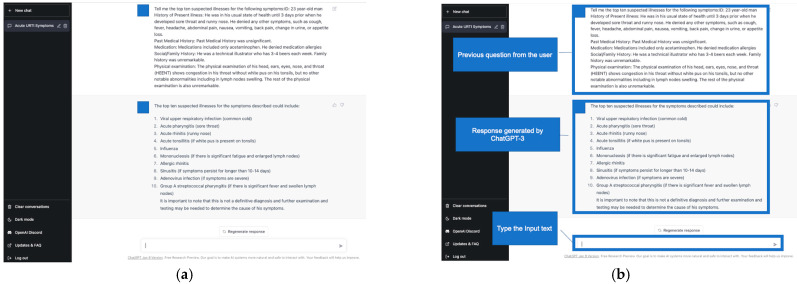
(**a**) Example of a differential-diagnosis list generated by ChatGPT-3. (**b**) Explanation of the differential-diagnosis list example generated by ChatGPT-3.

**Table 1 ijerph-20-03378-t001:** Rates of correct diagnoses within the ten and five differential-diagnosis lists, and as top diagnoses generated by ChatGPT-3. Rate of the correct diagnoses in the five differential diagnoses made by physicians.

Variable		ChatGPT-3			Physicians’ Diagnoses		*p* Values	
		Within Top 10	Within Top 5	As top Diagnoses	Within Top 5	As Top Diagnoses	Within Top 5 ^1^	As Top Diagnoses ^2^
total, *n*(%)		28/30 (93.3)	25/30 (83.3)	16/30 (53.3)	59/60 (98.3)	56/60 (93.3)	0.03	<0.001
1.	abdominal pain, *n*(%)	3/3 (100)	3/3 (100)	2/3 (66.7)	6/6 (100)	5/6 (83.3)	>0.99	>0.99
2.	fever, *n*(%)	3/3 (100)	3/3 (100)	2/3 (66.7)	6/6 (100)	5/6 (83.3)	>0.99	>0.99
3.	chest pain, *n*(%)	3/3 (100)	3/3 (100)	2/3 (66.7)	6/6 (100)	6/6 (100)	> 0.99	0.71
4.	breathing difficulty, *n*(%)	2/3 (66.7)	1/3 (33.3)	1/3 (33.3)	6/6 (100)	6/6 (100)	0.16	0.16
5.	joint pain, *n*(%)	3/3 (100)	3/3 (100)	2/3 (66.7)	6/6 (100)	6/6 (100)	>0.99	0.71
6.	vomiting, *n*(%)	3/3 (100)	1/3 (33.3)	0/3 (0)	6/6 (100)	6/6 (100)	0.16	0.02
7.	ataxia/difficulty walking, *n*(%)	3/3 (100)	3/3 (100)	2/3 (66.7)	6/6 (100)	6/6 (100)	>0.99	0.71
8.	back pain, *n*(%)	2/3 (66.7)	2/3 (66.7)	1/3 (33.3)	5/6 (83.3)	5/6 (83.3)	>0.99	0.45
9.	cough, *n*(%)	3/3 (100)	3/3 (100)	1/3 (33.3)	6/6 (100)	5/6 (83.3)	>0.99	0.45
10.	dizziness, *n*(%)	3/3 (100)	3/3 (100)	3/3 (100)	6/6 (100)	6/6 (100)	>0.99	>0.99

^1^ *p* values from chi-square scores compared with correct diagnosis by physicians within the five differential-diagnosis lists. ^2^ *p* values from chi-square scores compared with correct diagnosis by physicians within the five differential-diagnosis lists.

**Table 2 ijerph-20-03378-t002:** Rates of consistent differential diagnoses made by two independent physicians and those made within the ten lists by ChatGPT-3.

Variable		ChatGPT-3 ^1^	Consistent Differential Diagnoses by Two Physicians
total, *n*(%)		62/88 (70.5)	88/150 (58.7)
1.	abdominal pain, *n*(%)	7/9 (77.8)	9/15 (60.0)
2.	fever, *n*(%)	4/8 (50.0)	8/15 (53.3)
3.	chest pain, *n*(%)	9/11 (81.8)	11/15 (73.3)
4.	breathing difficulty, *n*(%)	7/8 (87.5)	8/15 (53.3)
5.	joint pain, *n*(%)	7/8 (87.5)	8/15 (53.3)
6.	vomiting, *n*(%)	6/9 (66.7)	9/15 (60.0)
7.	ataxia/difficulty walking, *n*(%)	6/10 (60.0)	10/15 (66.7)
8.	back pain, *n*(%)	5/7 (71.4)	7/15 (46.7)
9.	cough, *n*(%)	4/10 (40.0)	10/15 (66.7)
10.	dizziness, *n*(%)	7/8 (87.5)	8/15 (53.3)

^1^ The consistent differential diagnoses between the two physicians were used as the denominator.

## Data Availability

Not applicable.

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
