# Peer review of "Diagnostic Accuracy of Differential-Diagnosis Lists Generated by Generative Pretrained Transformer 3 Chatbot for Clinical Vignettes with Common Chief Complaints: A Pilot Study"

_ijerph, 2023, doi:10.3390/ijerph20043378_

Round 1
Reviewer 1 Report
Review of "Diagnostic Accuracy of Differential-diagnosis Lists Generated by Generative Pretrained Transformer 3 Chatbot for Clinical Vignettes with Common Chief Complaints: A Pilot Study"
Date: February 4, 2023
Through a pilot study, the authors evaluated the accuracy of differential-diagnosis lists generated by ChatGPT-3 for clinical vignettes with common chief complaints. The aim is to compare the diagnostic accuracy of differential diagnoses made by artificial intelligence (AI) chatbots and physicians. The research question is of particular importance that could pave the path for future diagnosis of diseases. The authors did an excellent job in presenting their work. The manuscript is very easy to read too. I don't have any major complaints except for a few minor ones, which I have listed below.
Comments:
1. In Table 1, we can't trust the p-values as they are based on a meager sample size of 3. Also, why did the authors not categorize the physicians' correct diagnosis counts in the same fashion they organized ChatGPT -3's counts (i.e., within top 10, within top 5, and top diagnosis)?
2. It needs to be clarified what the authors wanted to achieve from Table 2. Also, the denominators are different. It can be easily solved by explaining more in the text.
3. The Strengths and Limitations sections were very clearly written. However, I am concerned about the type of chief complaints and the correct diagnoses. The true diagnosis list includes less severe diseases such as URTI to more severe diseases such as myocardial infarction, pulmonary embolism, etc. Do the researchers believe that the severity of illness could impact the performance of the chatbots? Do they think that the very common, daily happening, less threatening diseases will be diagnosed with higher sensitivity and the same could decrease for more lethal, not easily diagnosable diseases? It can be checked by categorizing the diseases based on their severity (or rarity, to answer a different type of question). It could be beyond the scope of this work but could be a future direction that the researchers can highlight.
4. In the same direction as my previous comment, do the researchers think the accuracy could be affected by incomplete information fed to the chatbot? In the study, medical professionals developed the clinical vignettes. However, in the future, if a typical user must use the chatbot in an unsupervised fashion, the information fed to the chatbot could be incomplete (especially if a template/format is missing), resulting in highly erroneous output. Assessing the impact of the incompleteness of information was also something that I found missing in the manuscript.
Author Response
Dear Reviewer 1,
Thank you for your interest and valuable recommendations. Below, you will find the response to each of your comments.
Through a pilot study, the authors evaluated the accuracy of differential-diagnosis lists generated by ChatGPT-3 for clinical vignettes with common chief complaints. The aim is to compare the diagnostic accuracy of differential diagnoses made by artificial intelligence (AI) chatbots and physicians. The research question is of particular importance that could pave the path for future diagnosis of diseases. The authors did an excellent job in presenting their work. The manuscript is very easy to read too. I don't have any major complaints except for a few minor ones, which I have listed below.
Comments:
- In Table 1, we can't trust the p-values as they are based on a meager sample size of 3. Also, why did the authors not categorize the physicians' correct diagnosis counts in the same fashion they organized ChatGPT -3's counts (i.e., within top 10, within top 5, and top diagnosis)?
Response: As reviewer 1 pointed out, we categorized the physician’s correct diagnosis counts in the same fashion as that of ChatGPT-3 in Table1. In this study, the physicians' correct diagnosis counts within the top five and in the top diagnosis were created, but that of within top 10 was not created. That was why we could not add a top 10 physician’s correct diagnosis.
- It needs to be clarified what the authors wanted to achieve from Table 2. Also, the denominators are different. It can be easily solved by explaining more in the text.
Response: We thank the reviewer for this pertinent suggestion. As reviewer 1 pointed out, we added the explanation in the methods (lines 179–181) and results sections (lines 268-271).
- The Strengths and Limitations sections were very clearly written. However, I am concerned about the type of chief complaints and the correct diagnoses. The true diagnosis list includes less severe diseases such as URTI to more severe diseases such as myocardial infarction, pulmonary embolism, etc. Do the researchers believe that the severity of illness could impact the performance of the chatbots? Do they think that the very common, daily happening, less threatening diseases will be diagnosed with higher sensitivity and the same could decrease for more lethal, not easily diagnosable diseases? It can be checked by categorizing the diseases based on their severity (or rarity, to answer a different type of question). It could be beyond the scope of this work but could be a future direction that the researchers can highlight.
Response: As reviewer 1 pointed out, we did not evaluate the differential diagnoses as severity and rarity. However, these points were considered the future research directions. We clarified “clinically important” to add the severity or rarity in the limitations (lines 344-345).
- In the same direction as my previous comment, do the researchers think the accuracy could be affected by incomplete information fed to the chatbot? In the study, medical professionals developed the clinical vignettes. However, in the future, if a typical user must use the chatbot in an unsupervised fashion, the information fed to the chatbot could be incomplete (especially if a template/format is missing), resulting in highly erroneous output. Assessing the impact of the incompleteness of information was also something that I found missing in the manuscript.
Response: As reviewer 1 pointed out, the incompleteness of medical information was quite important for the general user. Therefore, we added another subheading 4.4 as risk for the general user (lines 364-382).
Thank you again for your valuable time and recommendations. We believe these corrections have improved the quality of the manuscript and we hope it is worthy of publication.
Please contact us if any of our responses have not satisfied you.
Sincerely,
Takanobu Hirosawa, MD, PhD
Dokkyo Medical University Hospital, Kitakobayashi 880, Shimotsuga, Japan
Telephone: + 81-282-87-2498; fax: + 81-282-87-2502
e-mail: hirosawa@dokkyomed.ac.jp
Reviewer 2 Report
Dear Authors,
Thank you for submitting your work to the international journal of Environmental Research and public health – MDPI. The idea is discussed successfully, and it’s worthily investigated. However, some improvements need to be considered to make your work better.
General comment:
The paper structure is a bit confusing, and an important section should be added (related work/ literature review).
Detailed comments:
1. Introduction
· I suggest adding a paragraph consisting of two to three sentences to describe this part before starting to discuss the sub-headings 1.1, 1.2..etc.
· I suggest adding a paragraph representing a paper map which describes what are the things that going to be discussed in the rest of the paper.
· The author should add a new part that discusses related work or a literature review
2. Materials and Methods
· I suggest adding a paragraph consisting of two to three sentences to describe this part before starting to discuss the sub-headings 2.1, 2.2..etc.
4. Discussion
· I suggest adding a paragraph consisting of two to three sentences to describe this part before discussing the sub-headings 4.1, 4.2..etc.
· 4.4. Comparison with Prior Work
This part is ok, but I suggest transferring it to the suggested related work/ literature review (to be in section 2) and providing a brief about each prior work.
Thank you
Author Response
Dear Reviewer 2,
Thank you for your interest and valuable recommendations. Below you will find the response to each of your comments.
Thank you for submitting your work to the international journal of Environmental Research and public health – MDPI. The idea is discussed successfully, and it’s worthily investigated. However, some improvements need to be considered to make your work better.
General comment:
The paper structure is a bit confusing, and an important section should be added (related work/ literature review).
Detailed comments:
- Introduction
- I suggest adding a paragraph consisting of two to three sentences to describe this part before starting to discuss the sub-headings 1.1, 1.2..etc.
Response: As reviewer 2 pointed out, we added several sentences before the subheadings (lines 29-33).
- I suggest adding a paragraph representing a paper map which describes what are the things that going to be discussed in the rest of the paper.
Response: As reviewer 2 pointed out, we added a paragraph representing a paper map (lines 106-117).
- The author should add a new part that discusses related work or a literature review
Response: As reviewer 2 pointed out, we added a new part, Related Work, as 1.4 (lines 95-101).
- Materials and Methods
- I suggest adding a paragraph consisting of two to three sentences to describe this part before starting to discuss the sub-headings 2.1, 2.2..etc.
Response: As reviewer 2 pointed out, we added several sentences before the subheadings (lines 119-121).
- Discussion
- I suggest adding a paragraph consisting of two to three sentences to describe this part before discussing the sub-headings 4.1, 4.2..etc.
Response: As reviewer 2 pointed out, we added several sentences before the subheadings (lines 283-284).
- 4.4. Comparison with Prior Work
This part is ok, but I suggest transferring it to the suggested related work/ literature review (to be in section 2) and providing a brief about each prior work.
Response: As reviewer 2 pointed out, we provided a brief description about the prior work in related works or a literature review(1.4, lines 95-101). We also transferred a part of this section (lines 82-88). The “comparison with prior work” section related to the current result would be friendly for readers after the results section. That is why we did not transfer everything.
Thank you again for your valuable time and recommendations. We believe these corrections have improved the quality of the manuscript and we hope it is worthy of publication.
Please contact us if any of our responses have not satisfied you.
Sincerely,
Takanobu Hirosawa, MD, PhD
Dokkyo Medical University Hospital, Kitakobayashi 880, Shimotsuga, Japan
Telephone: + 81-282-87-2498; fax: + 81-282-87-2502
e-mail: hirosawa@dokkyomed.ac.jp